# Molecularly Designed Ion-Imprinted Nanoparticles for Real-Time Sensing of Cu(II) Ions Using Quartz Crystal Microbalance

**DOI:** 10.3390/biomimetics7040191

**Published:** 2022-11-05

**Authors:** Nihan Aydoğan, Gülgün Aylaz, Monireh Bakhshpour, Tugba Tugsuz, Müge Andaç

**Affiliations:** 1Department of Environmental Engineering, Environmental Technology Division, Hacettepe University, Beytepe, Ankara 06800, Turkey; 2Nanotechnology and Nanomedicine Division, Hacettepe University, Beytepe, Ankara 06800, Turkey; 3Nanotechnology Engineering Department, Sivas Cumhuriyet University, Sivas 58140, Turkey; 4Department of Chemistry, Biochemistry Division, Hacettepe University, Beytepe, Ankara 06800, Turkey; 5Department of Chemistry, Theoretical Chemistry Division, Hacettepe University, Beytepe, Ankara 06800, Turkey

**Keywords:** molecularly designed ion-imprinted nanoparticles, gravimetric nanosensor, quartz crystal microbalance (QCM), Cu(II) ions

## Abstract

A molecularly designed imprinting method was combined with a gravimetric nanosensor for the real-time detection Cu(II) ions in aqueous solutions without using expensive laboratory devices. Thus, 1:1 and 2:1 mol-ratio-dependent coordination modes between Cu(II), N-methacyloly-L histidine methyl ester (MAH) functional monomer complexes, and their four-fold and six-fold coordinations were calculated by means of density functional theory molecular modeling. Cu(II)-MIP1 and Cu(II)-MIP2 nanoparticles were synthesized in the size range of 80–100 nm and characterized by SEM, AFM and FTIR. Cu(II)-MIP nanoparticles were then conducted to a quartz crystal microbalance sensor for the real-time detection of Cu(II) ions in aqueous solutions. The effects of initial Cu(II) concentration, selectivity, and imprinting efficiency were investigated for the optimization of the nanosensor. Linearity of 99% was obtained in the Cu(II) ion linear concentration range of 0.15–1.57 µM with high sensitivity. The LOD was obtained as 40.7 nM for Cu(II)-MIP2 nanoparticles. The selectivity and the imprinting efficiency of the QCM nanosensor were obtained significantly in the presence of competitive ion samples (Co(II), Ni(II), Zn(II), and Fe(II)). The results are promising for sensing Cu(II) ions as environmental toxicants in water by combining molecularly designed ion-imprinted nanoparticles and a gravimetric sensor.

## 1. Introduction

As a result of development in technology and industry, contamination stresses on air, water, and soil environments are becoming extremely high. In addition, the adaptation of living organisms to this situation cannot be realized at the same rate. Many living organisms, particularly aquatic animals, plants, and humans, are exposed to industrial pollutants, and these pollutants have started to change the ecosystem. In order to reduce the impact of toxic pollutants on living organisms that is caused by industry, research is proceeding toward a new, reliable, and reproducible system that can be analyzed without the need for laboratory conditions. Although it is an important micronutrient for humans, copper (Cu) is highly toxic at an elevated level. Cu compounds are added to fertilizers and animal feed as nutrients that support plant and animal growth. They are also used as food additives (e.g., nutrients or coloring agents). Copper sulfate pentahydrate is sometimes added to surface water to control algae [1]. By attaching to the thiol groups in copper proteins or by triggering Fenton-type processes, it generates reactive hydroxyl radicals [2]. Such chelates can facilitate the relatively irregular transfer of electrons to macromolecules, such as lipids, proteins, and DNA, if metal ions are sometimes allowed to exist in relatively unstructured chelates consisting of mono- or biaxial ligands (sometimes called ‘free’ in a biological system) [3]. Cells regulate the traffic of the transition essence ions and maintain the quantity necessary for natural function by avoiding extreme situations that are toxic. Injured tissues deliver the raised radical-generating enzyme activation of phagocytes, as well as release free iron or copper ions and disrupt the electron transport chains of oxidative phosphorylation. The transfer of copper takes place directly from protein to protein, thereby heading off radical products [2].

The overloading of copper leads to Fenton-type redox reactions, causing oxidative cell damage and cell death [4]. Cu(II) can react with molecular oxygen to produce reactive oxygen species (ROS) and damage biomolecules at high concentrations. In this context, the level of Cu(II) residue in drinking water is limited. The United States Environmental Protection Agency (USEPA) determined the maximum pollutant level (MCL) in drinking water at 1300 µg/L Cu(II), while the European Commission (EC) set the limit at 2000 µg/L, with the World Health Organization (WHO) also at 2000 µg/L. Therefore, it is important to develop sensitive and selective analytical methods for the detection of Cu(II) ions [1,5].

Among the most important analytical methods used for the detection of copper in water are flammable atomic absorption spectrophotometry (FAAS), graphite furnace atomic absorption spectroscopy (GFAAS), inductively coupled plasma atomic emission spectroscopy (ICP-AES), and inductively coupled plasma mass spectrometry (ICP-MS). More specifically, there are the methods of inductively coupled plasma optical emission spectrometry (ICP-OES) and X-ray fluorescence (XRF) [6]. While the ICP-MS technique has the lowest (0.02 μg/L) and the AAS technique has the highest (20 μg/L) detection limit for copper, the other technique detection levels range from 0.7 to 3 μg/L. For these analytical methods, the measurement of dissolved copper requires prefiltration. Results from unfiltered samples include dissolved and particulate copper. Simple colorimetric methods are also applicable for copper measurement, but the method sensitivity is very low. The detection limit for copper is determined as 50 μg/L by colorimetric methods [1,7,8,9].

Molecular-imprinting technology is one of the most promising technologies in recognition and detection systems due to its high selectivity, good stability, simplicity, and low cost. Molecularly imprinted polymeric materials are mainly use to design artificial biomimetic receptors for the selective removal and preconcentration of imprinted analytes [10,11,12,13]. Molecularly imprinted polymers are highly stable structures with recognition regions (cavities) within a polymer matrix adapted to the three-dimensional (3D) shape and functions of an analyte. The analyte-specific cavities on imprinted materials can be produced in a covalent or non-covalent way [14].

With molecular-imprinting technology, small molecules, as well as large molecules, can be detected in sensing applications. Molecular-imprinting technology is based on mimicking natural biological interactions and binding molecules thanks to polymers, which are developed by organizing functional monomers around a template. Imprinting at the molecular level is a biomimetic strategy that allows synthetic or semi-synthetic materials to simulate the physical and chemical structures seen in natural systems [15]. In ion sensing, various ion-imprinted polymer (IIP)-based sensors have been developed for the detection of toxic metal ions. Among them, the quartz crystal microbalance (QCM) method, a high-resolution mass-sensing technique that measures mass changes in a vibrating quartz crystal surface in real time, is quite advantageous in obtaining unlabeled, selective, simple, and stable sensing. The QCM technique has been one of the frequently used methods for detecting environmental pollutants [16,17].

In this study, QCM sensor activity is examined for the detection of Cu(II) ions using a molecularly designed imprinting method. The interaction between Cu(II) ions and the functional monomer used in polymerization is virtually analyzed with computational methods. The exact stable structures of the Cu(II) and complexes are crucial for understanding the chemical mechanism of their imprinting activity. Thus, the coordination of 1:1 and 2:1 mol ratios of Cu(II) and MAH complexes additionally coordinated with H2O molecules are calculated by means of DFT at the BP86/6-31G (d, p) and LanL2DZ levels. The most stable conformation of Cu(II) ion in the complexes is selected for the imprinting mechanism. The functional monomer MAH is used as the metal-coordinating ligand owing to the affinity of nitrogen donor atoms located in the imidazole group of histidine residue toward metal ions for the separation of various analytes [18]. Cu(II)-imprinted (Cu(II)-MIP) nanoparticles are conducted to the QCM nanosensor for the real-time sensing of Cu(II) ions in aqueous solutions. The binding parameters of the Cu(II)-MIP nanoparticles, such as the effects of Cu(II) ion initial concentration, selectivity, and imprinting efficiency in the presence of competitive ions, are optimized through QCM nanosensor studies.

## 2. Materials and Methods

### 2.1. Materials

The monomers 2-hydroxyethyl methacrylate (HEMA) and ethylene glycol dimethacrylate (EGDMA), the stabilizer polyvinyl alcohol (PVAL), the surfactant sodium dodecyl sulfate (SDS), and the initiators sodium bisulfite (NaHSO_3_) and ammonium persulfate (APS) were obtained from Sigma (St. Louis, MA, USA) for the synthesis of nanoparticles. All other chemicals used in the experiments were of analytical purity. The specific functional monomer N-Methacryloyl-L-histidine methyl ester (MAH) was supplied from our collaborator research group. In addition, the deionized water used was purified (18 μS/cm) using a Barnstead D3804 NANOpure^®^ (Thermo Fisher Scientific, Shelton, CT, USA) brand organic and colloid removal unit and an ion exchange filled column system. The glass materials used in the experiments were kept in a 5% nitric acid solution overnight and dried in a dust-free environment by rinsing with deionized water before use.

### 2.2. Geometries of Cu(II)-MAH Complexes

The computational calculations were carried out using the Gaussian 09 Program [19]. The generalized gradient approximation (GGA) of BP86 functional, which was composed of Becke 1988 exchange functional and Perdew 86 correlation functional, was used [20]. The double ζ″ quality basis set of 6-31G (d, p) for C, N, O, and H atoms [21] and the LANL2DZ basis set for Cu atoms [22] were used for optimization. Complexes with +2 charges were optimized without any symmetry restrictions in the doublet state. Frequency calculations for the complexes produced no imaginary frequencies, indicating that in each case the minimum energy point was located.

The binding energies (EB) of the 1:1 and 2:1 Cu(II)-MAH water complexes were calculated with the following equation:(1)$$E_{B}=E_{\mathrm{Complex}}-\left[\left(E_{\mathrm{Cu}\left(\mathrm{II}\right)}+n\times E_{\mathrm{MAH}}+n\times E_{\mathrm{aq}}\right)\right]$$

Here, E_B_ is the binding energy; E_Complex_ is the total energy of the complex; n is the number of constituent species; and E_Cu_, E_MAH_, and E_aq_ are the energies of Cu(II), MAH, and water, respectively.

In experimental studies, Cu(II) ions were complexed with the MAH monomer in two different mole ratios. Mainly, MAH monomer and Cu(II) ions in 1:1 and 2:1 mol ratios were allowed to incubate in 1 mL of aqueous solution under slight stirring for 2 h at room temperature. The prepared monomer-metal ion complexes were used fresh in the polymerization process. The chemical structures of the MAH-Cu(II) complexes were characterized using attenuated total reflection-Fourier transform infrared (ATR-FTIR) spectroscopy (Thermo Fisher Scientific, Nicolet iS, Waltham, MA, USA). The prepared MAH-Cu(II) precomplexes were used fresh in the synthesis of Cu(II)-MIP nanoparticles.

### 2.3. Synthesis and Characterization of Cu(II)-MIP and NIP Nanoparticles

MIP nanoparticles were prepared in two different groups according to the mole ratio of the MAH-Cu(II) complexes. The first MIP nanoparticle group (MIP1) was synthesized in the presence of 1:1 mmol of MAH-Cu(II), with the second group in 2:1 mmol of MAH-Cu(II) complex. Non-imprinted (NIP) nanoparticles were synthesized without using Cu(II) ions for the control experiments. The emulsion polymerization method was used for the synthesis of all MIP and NIP nanoparticles. Accordingly, two phases of the aqueous emulsion medium, Phase I (10 mL water-dissolved 0.187 g PVA) and Phase II (150 mL water-dissolved 0.1 g PVA and 0.1 g SDS), were prepared. The organic phase, which contained HEMA (0.4 mL) and EGDMA (2.1 mL) as monomer and cross-linker, respectively, and 1 mL of predesigned MAH-Cu(II) complex were carefully added to Phase I and homogenized at 10,000 rpm for 5 min. The resulting mini-emulsion medium was slowly added to the Phase II aqueous medium under 600 rpm stirring at 40 °C in a continuous-mixing reactor. Subsequently, the initiators NaHSO3 and APS (1% *w*/*w*) were added, and the polymerization process was carried out at 40 °C for 24 h. When the polymerization was completed, the nanoparticles were firstly washed with ethanol solution (25% *v*/*v*) to remove the surfactant and unreacted monomers, and then nanoparticles were further washed with EDTA solution (pH 4.0, 50 mM) to remove Cu(II) ions. An ATR-FTIR spectrometer (Thermo Fisher Scientific, Nicolet iS, Waltham, MA, USA) was used in the range of 600–4000 cm^−1^ for structural characterization of MIP nanoparticles. Sizes of the suspended MIP nanoparticles (1 mg/mL) in DI water were measured with a Nano Zetasizer (NanoS, Malvern Instruments, London, UK).

### 2.4. Conduction of Cu(II)-MIP and NIP Nanoparticles to QCM Nanosensor

The QCM sensor surface was washed with piranha solution (1:4 H_2_O_2_:H_2_SO_4_ (*v*/*v*)) to remove any contaminants and then dried at 40 °C in an oven. After the cleaning process, the suspended (1 µg/µL) MIP1, MIP2, and NIP nanoparticles were separately applied to QCM sensors, and they were cured under UV light for 30 min. Surface thickness measurements with an ellipsometer (Nanofilm EP3-Nulling Ellipsometer, Gottingen, Germany) were carried out at a wavelength of 532 nm and an angle of 62°. They were repeated 3 times at 6 different points on the nanosensor surface, and the average values were reported. The contact angle measurements (KRÜSS DSA100, Hamburg, Germany) were obtained using the adherent drop (Sessile Drop) method [23] with data from 10 repeated tests. Structural-imaging studies of the nanoparticles were carried out using an atomic force microscope (AFM, Nanomagnetics Instruments, Oxford, UK) in tapping mode and an air atmosphere. The oscillation resonance frequency was set as 341.30 kHz. The nanoparticles on the sensor surface were scanned at a 2 µm/s rate on a 2 × 2 µm^2^ sample area [24]. The nanoparticles were frozen at −20 °C and dried in a lyophilizer (Chris Alpha 1-2 LD plus, M Christ GmbH, Germany) prior to scanning electron microscopy (SEM, JEOL, JEM 1200 EX, Tokyo, Japan) studies. The dried MIP nanoparticle samples were mounted on stubs fitted with adhesive carbon pads, coated with Au–Pd (40:60), and visualized using SEM.

### 2.5. MIP Efficiency of QCM Nanosensor

MIP-QCM nanosensing was conducted with a QCM device (RQCM, Inficon Acquires Maxtek Inc., CA California, USA) and used for the detection of Cu(II) ions in aqueous solutions. Cu(II) solutions with different initial concentrations in the range of 0.01–100 µg/mL were prepared in 0.1 M acetate buffer (pH 4.0). Firstly, the QCM nanosensor surface was rinsed with deionized water (50 mL), and then the frequency was equilibrated with 0.1 M pH 4.0 acetate buffer at room temperature (for nearly 3 min). The resonance frequency shift (∆f) was recorded simultaneously about 5 min after the feed of Cu(II) solutions of each concentration. Once it reached equilibrium, the bound Cu(II) ions were desorbed with 50 mM of EDTA solution. The MIP-QCM nanosensor was repeatedly rinsed with deionized water and binding buffer for each cycle. The resonance frequency change was evaluated using RQCM Maxtek software. A fine-oscillation quartz crystal microbalance compressed between two metal excitation electrodes was used to determine the mass changes on the interface during the mass dependence of the QCM resonance frequency. The relationships between the shift in resonance frequency (Δf) and the mass change (Δm) of adsorbed and desorbed Cu(II) ions were determined according to the Sauerbrey equation (Equation (2)). Using the linear connection between mass change (Δm) and resonance frequency change (Δf), a calibration graph was created with the standard Cu(II) solution concentrations.
(2)$$\Delta m=-C_{Qcm}\times \Delta f$$

Here, Δm denotes the mass change, and Δf denotes the frequency shift. The mass sensitivity constant (C_Qcm_) equals to 17.7 ng/cm^2^.Hz for 5 MHz AT cut of QCM sensor.

In order to investigate the reusability of the MIP-QCM nanosensor, repeated adsorption and desorption studies were conducted 5 times consecutively with 1 µg/mL of Cu(II) ion solution. The QCM sensor surface containing Cu(II)-MIP nanoparticles was stabilized with pH 4 acetate buffer for 3 min, and then treated with 1 µg/mL Cu(II) solution for 5 min. Then, Cu(II) ions were desorbed from the nanosensor with 50 mM EDTA for 2 min. In each cycle, it was evaluated whether the system reached the same frequency value. The changing amounts of resonance frequency were recorded with the Cu(II) adsorption and desorption steps using real-time monitoring. The QCM nanosensor system was tested with reusable cycles until an insignificant decrease or increase was observed.

### 2.6. Equilibrium Isotherm Models, Detection, and Quantification Limits

Three different equilibrium isotherm models were applied to evaluate the interaction mechanism between the Cu(II) ions and the MIP-QCM nanosensors. These were the Langmuir, Freundlich, and Langmuir–Freundlich models. The equations for these equilibrium isotherm models are given in Equations (3)–(6):(3)$$\mathrm{Langmuir}:\;\Delta m=\left(\Delta m_{max}c/K_{D}+c\right)$$
(4)Freundlich: Δm=Δmmaxc1n
(5)Langmuir–Freundlich: Δm=(Δmmaxc1n/KD+c1n)
(6)$$K_{A}=1/K_{D}$$

Here, Δm_max_ is the maximum QCM mass change; Δm is the equilibrium QCM mass change; c is the analyte concentration (nM); K_A_ (1/nM) is the association constant; K_D_ (nM) is the dissociation constant; and 1/n refers to the Freundlich surface heterogeneity index.

The sensitivity of the MIP-QCM nanosensors was confirmed with standard Cu(II) solutions. The confidence interval was taken as 95%, and the data were obtained from experiments performed 3 times. The minimum concentration or the lowest amount of substance that could be detected by a particular method within a specified confidence cutoff was the limit of detection (LOD). The limit of quantification (LOQ) was the level of linearity at which the analyte could be quantified with an acceptable degree of precision and accuracy, constituting the lowest concentration. The LOD and LOQ of the Cu(II)-MIP QCM nanosensors were calculated using the following equations:(7)$$LOD=3.3x\;\left(\frac{s}{m}\right)$$
(8)$$LOQ=10x\left(\frac{s}{m}\right)$$

Here, the s value represents the standard deviation, and the m value represents the slope of the calibration curve.

### 2.7. Selectivity and Imprinting Efficiency of MIP-QCM Nanosensors

The selectivity experiments were performed in the presence of competitive ion samples, such as nickel (Ni(II)), cobalt (Co(II)), zinc (Zn(II)), and iron (Fe(II)). The relationships between the shift in resonance frequency (Δf) and the mass change (Δm) of the competitor ions adsorbed and desorbed were determined according to the Sauerbrey equation (Equation (2)). The selectivity (K) of the Cu(II) ions according to the competitor ions were calculated with Equation (9):(9)$$K=\Delta m_{\mathrm{Cu}\left(\mathrm{II}\right)}/\Delta m_{\mathrm{Competitors}}$$

Here, K is the selectivity, and Δm is the equilibrium QCM mass changes for Cu(II) ions and competitor ions.

In addition, the imprinting efficiencies (K′) of the MIP1- and MIP2-QCM nanosensors with regard to the NIP-QCM nanosensor were calculated using Equation (10):(10)$$K^{\prime }=K_{MIP}/K_{NIP}$$

Here, K′ is the imprinting efficiency, and K is the selectivity of MIP nanoparticles.

## 3. Results and Discussion

### 3.1. Characterization of Molecularly Designed Complexes and MIP Nanoparticles

Transition metal ions, such as Cu(II), Ni(II), Zn(II), and Co(II), which are known as Lewis acids and are considered electron pairs, can coordinate with electron-donating groups (such as N, S, and O) in chelating compounds. The remaining metal coordination zones are normally retained by water molecules [25]. N-methacryloyl-L-histidine methyl ester (MAH) was developed as an affinity-chelating monomer for the separation of different kinds of analytes by introducing its functional histidine group into the polymeric structure without any ligand leakage [26,27]. Concisely, MAH is an L-histidine-derived polymerizable functional monomer and has recently been used as in the synthesis of molecularly imprinted polymers for metal ion separation and water treatment studies [28,29]. It is capable of coordinating with transition metal ions through its imidazole residue, which is located in the side chain of the histidine amino acid. In this study, the interactions between the MAH monomer and Cu(II) ions were investigated through both computational and analytical tools. The MAH monomer was used to form complexes with Cu(II) ions in two different mole ratios, mainly 1:1 (MAH-Cu(II)) and 2:1 (2MAH-Cu(II)). The geometry optimizations of MAH and its 1:1 and 2:1 complexes of four-fold and six-fold Cu(II)-MAH structures were studied with computational modeling (Figure 1). The structures of MAH and its complexes were calculated in gas phase using the BP86/6-31G(d, p) and BP86/6-31G(d, p)+LanL2DZ levels of theory, respectively. The calculated total and binding energies are given in Table 1. Figure 1a shows the stable conformer of keto tautomer in the MAH molecule. Due to the all the calculation results, the most stable form was found as MAH-3-keto. In the enol forms, the most stable form was calculated as MAH-3-enol (Appendix A).

From the calculation results of the four-fold and six-fold MAH-Cu(II) complexes, the most stable structure was the enol form of the complexes in which copper(II) was bonded with the nitrogen atom of the histidine ring, including three and five water molecules, respectively (Figure 1b,c). In this most stable structure, Cu(II) coordinated with MAH and water in its distorted square planar and distorted trigonal biplanar structures for four-fold and six-fold coordination, respectively. In the complexation with both N and O atoms of MAH and Cu(II), square planar structures were found for four-fold with two water molecules, as well as square pyramid or trigonal bipyramidal structures for five-fold with three water molecules (Appendix A).

In our experimental results, we found that Cu(II) could coordinate with two MAH monomers (Figure 2). Thus, we calculated Cu(II) coordination with two MAH monomers and water due to their four-fold and six-fold structures. Figure 1d and 1e show the four-fold and six-fold coordinations of Cu(II)-(MAH)_2_ complexes. Due to the binding energies of the four- and six-fold coordinated complexes in Table 1, the Cu(II)-(MAH)_2_ complexes of both conformers were found as more stable (with −0.7114 and −0.7655 Hartree energies, respectively) than Cu(II)-(MAH) complexes, which was in good agreement with the experimental results.

The structural characterizations of the MAH monomer and MAH-Cu(II) complexes were also confirmed experimentally. The resulting ATR-FTIR spectra of the MAH monomer (upper side), MAH-Cu(II) (middle), and 2MAH-Cu(II) (bottom side) complexes are shown in Figure 2. The characteristic peaks of the MAH monomer in Figure 2 (upper) were assigned to OH (3354 cm^−1^), C-C strain (2997, 2952, and 2927 cm^−1^), and C = O strain (1743 cm^−1^). The presence of characteristic amide peaks of the MAH monomer as amide I, amide II, and amide III were observed at 1676, 1516, and 1437 cm^−1^, respectively (red circles in Figure 2). At these frequency ranges (in circles), the characteristic amide peaks of the MAH-Cu(II) (middle) and 2MAH-Cu(II) (bottom) complexes were broadened and shifted to lower frequencies due to the metal coordination complex of MAH and the Cu(II) ions. In addition, the 2MAH-Cu(II) (bottom side) complex indicated an intense broadening. This observation could be explained by the more intense donation of electrons from the lone pair of nitrogen located in the imidazole residue to the empty d-orbital of the Cu(II) ions.

Prior to Cu(II)-MIP nanoparticle preparation, the Cu(II) ions were used as a template to form a complex with the MAH monomer, the functional monomer. The polymerization step took place in the presence of a cross-linker (EGDMA), a basic monomer (HEMA), and a functional monomer template complex (MAH-Cu(II)). After polymerization, the template Cu(II) ions were removed. In this way, Cu(II) ions left their places in the imprinted cavities (Figure 1) in structure and shape memory.

The NIP nanoparticles were used as a control in the selectivity experiments, and they were synthesized without template Cu(II) ions. The structural and surface features of the Cu(II)-MIP nanoparticles and the Cu(II)-MIP QCM nanosensor were examined, respectively.

FTIR analyses were conducted for the structural characterization of Cu(II)-MIP1 and MIP2 nanoparticles. The characteristic peaks of Cu(II)-MIP1 and MIP2 nanoparticles observed in amide I (1643 and 1626 cm^−1^), amide II (1450 and 1453 cm^−1^), amide III (1386 and 1389 cm^−1^) belonged to the imidazole groups of OH (3526 and 3512 cm^−1^), C-C strain (2946 and 2950 cm^−1^), and C = O strain (1723 and 1723 cm^−1^) respectively. The Cu(II)-MIP1 and MIP2 nanoparticles did not contain amide groups other than the MAH monomer. Therefore, the presence of amide groups seen in the band range of 1600–1400 cm^−1^ confirmed that MAH groups entered the structure of the Cu(II)-MIP1 and MIP2 nanoparticles (Appendix A).

A Zeta Sizer was used to determine the size distribution of the Cu(II)-MIP1 and MIP2 nanoparticles. According to the results, the sizes of the MIP nanoparticles in 1 mg/mL aqueous solutions were seen in the range of 100–150 nm. As the nanoparticles were in the aqueous solution for size analysis with the Zeta Sizer, the nanoparticles were swollen. This bloating caused the particles to change in size in the range of 20–50 nm.

### 3.2. Conduction of Cu(II)-MIP Nanoparticles to QCM Nanosensor

Surface characterizations of the Cu(II)-MIP QCM nanosensors were performed via ellipsometry, contact angle analysis, AFM, and SEM. The average thickness values of the Cu(II)-MIP1 and MIP2 QCM nanosensors were calculated as 15.0 ± 0.04 nm and 12.0 ± 0.04 nm, respectively (Figure 3).

Figure 4a,b show the contact angles on the surfaces of the QCM nanosensors, which contained the Cu(II)-MIP1 and MIP2 nanoparticles at molar ratios of 1:1 and 2:1, respectively. While the contact angle on the QCM nanosensor surface was measured as 46.9 θ° with Cu(II)-MIP1 nanoparticles in a 2:1 molar ratio, the contact angle was 41.8 θ° on the surface of the QCM nanosensor containing the 1:1 molar ratio of Cu(II)-MIP1 nanoparticles (Figure 4). Table 2 shows the contact angle values on the QCM nanosensor surfaces without nanoparticles, as well as containing Cu(II)-MIP1 and MIP2 nanoparticles. The contact angle values on the QCM nanosensors containing Cu(II)-MIP1 and MIP2 nanoparticles decreased considerably compared to that without nanoparticles on the nanosensor surface. It was seen that the Cu(II)-MIP1 and MIP2 QCM nanosensor surfaces showed that the HEMA monomer was highly hydrophilic due to the rich hydroxyl (OH) groups, so the desired nanoparticles were present on the nanosensor.

Figure 5a gives a three-dimensional AFM image of Cu(II)-MIP2 nanoparticles in tapping mode The surface depth of the bare QCM nanosensor was 10 nm. According to the results, the average of the Cu(II)-MIP2 nanoparticles was 81 nm. The presence of Cu(II)-MIP2 nanoparticles on the QCM nanosensor surface was clearly visible.

The surface structures of the Cu(II)-MIP1 and MIP2 nanoparticles and their average diameters were visualized with SEM. Figure 5a–c show the surface morphologies of Cu(II)-MIP1 and MIP2 nanoparticles, respectively. According to the SEM images, the average size of the Cu(II)-MIP nanoparticles varied between 80 nm and 120 nm. These results were consistent with the results from the AFM and Zeta size distribution measurements.

### 3.3. Efficiency of Cu(II)-MIP-Based QCM Nanosensor

Cu(II)-MIP1 and MIP2 nanoparticles prepared in two different molar ratios were adapted to a QCM nanosensor system. Figure 6 shows the time–Δf change and time–Δm change of Cu(II) solutions applied at different concentrations of Cu(II)-MIP1 and MIP2 nanoparticles, respectively. As it can be seen from the figures, with the binding of the mass to the quartz surface, a decrease in the frequency of crystalline oscillation, as well as a significant increase in mass awareness, were observed.

The Δm value of the Cu(II)-MIP1 and MIP2 QCM nanosensors increased as the concentration of Cu(II) ions increased. The Δm value of the Cu(II)-MIP1 QCM nanosensor reached about 15 µM, and the Δm value of the Cu(II)-MIP2 QCM nanosensor reached equilibrium at approximately 78 µM (Figure 7). At the end of five repeated rebinding cycles of the Cu(II)-MIP QCM nanosensor, no significant decrease in the binding capacity of Cu(II) was observed. This result shows that the Cu(II)-MIP QCM nanosensors could be used repeatedly while maintaining the Cu(II) binding capacity.

### 3.4. Equilibrium Isotherm Models

The Langmuir adsorption model is based on a homogeneous binding assumption, while the Freundlich adsorption model is based on a heterogeneous binding assumption. The Langmuir model has been extensively used for binding isotherms using molecularly imprinted polymers [30]. The Langmuir model generally refers to monolayer adsorption behavior. It was recently reported that molecularly imprinted polymers also contained heterogeneous binding sites [31]. The Freundlich adsorption model can be suitable for MIP systems, especially at low concentrations. However, this model shows some deviations in high-concentration values. To prevent these deviations, the Langmuir–Freundlich binary model can be used. This model is compatible with MIP systems, from very low concentrations to saturation, where there is heterogeneity [32]. The Cu(II)-MIP1 QCM nanosensor results obtained from all three isotherm models are summarized in Table 3.

The linearity of the adsorption models calculated with the experimental data for the Cu(II)-MIP1 and Cu(II)-MIP2 QCM nanosensors was examined. While it was seen that the Langmuir–Freundlich adsorption model (R^2^ = 0.99) gave more compatible results for Cu(II)-MIP1, the Langmuir adsorption model (R^2^ = 0.99) was more compatible for the Cu(II)-MIP2 QCM nanosensor. In addition, for the both QCM nanosensors, the Δm_max_ values (0.309 and 0.628) obtained from the experimental graph were the closest results to the theoretical Δm_max_ values (0.292 and 0.606) calculated in the Langmuir adsorption model. These results indicate that the Cu(II)-MIP1 QCM nanosensor’s Cu(II)-binding sites were both homogeneous and heterogeneously distributed at both high and low concentrations, which were frequently observed in MIPs [33]. In addition, theoretically calculated K_A_ and K_D_ values from the Langmuir–Freundlich equation were calculated as 0.168 µM^−1^ and 5.945 µM, respectively. On the other hand, the Cu(II)-MIP2 QCM nanosensor chip surface was homogeneously distributed, monolayer, and equipotential. The theoretically calculated K_A_ and K_D_ values from the Langmuir equation were calculated as 0.887 µM^−1^ and 1.127 µM, respectively.

### 3.5. Detection and Quantification Limits of Cu(II)-MIP QCM Nanosensor

Method validation was performed by the real-time sensing of Cu(II) ions with the Cu(II)-MIP QCM nanosensor. The linearity (R^2^ value) of the data in the range of 0.15–1.57 µM of the Cu(II)-MIP1 and MIP2 QCM nanosensors was calculated as 0.99. According to these results, Cu(II)-MIP1 and MIP2 QCM nanosensors operated with high affinity and 99% linearity in the concentration range of 0.15–1.57 µM. The linearity interval data and calculated LOD and LOQ values are given in Table 4.

### 3.6. Selectivity of Cu(II)-MIP QCM Nanosensors

When the Cu(II)-MIP1 and Cu(II)-MIP2 QCM nanosensors were compared with the competitive ions of Co(II), Ni(II), Fe(II), and Zn(II), it was observed that the Δm value of the target Cu(II) ions was higher than those of the competitive metal ions (Figure 8). Although the ion diameters and charges of the selected metals were similar, MIP1 and MIP2 show high selectivity to Cu ions.

The Δm values of both Cu(II)-MIP1 and Cu(II)-MIP2 nanosensors denoted that the Cu(II) ions’ selectivity (K) of Cu(II)-MIP nanoparticles were significantly higher than those of the competitive metal ion samples (Co(II), Ni(II), Fe(II), and Zn(II)). The results showed that Cu(II) ions, with respect to Co(II), exhibited the highest selectivity at 8.1 and 6.9 for the Cu(II)-MIP1 and Cu(II)-MIP2 nanosensors, respectively. Moreover, the imprinting efficiencies (K′) of the Cu(II)-MIP nanoparticles with respect to NIP nanoparticles were substantially effective in the presence of the competitive ion samples. The selectivity (K) and imprinting efficiency (K′) values of the Cu(II)-MIP nanoparticles are listed in Table 5.

There are limited studies on Cu(II)-imprinted polymeric particles in the literature, but a few studies that have detected Cu(II) with different sensor systems and adsorption mechanisms are summarized in Table 6. Many methods, such as fluorescence spectroscopy, voltammetric measurements, UV-vis absorption, atomic absorption, and ICP emission spectroscopy, are used to detect metal ions. Although interest in fluorescent sensors has increased in recent years, there are some problems encountered in the detection of metal ions. Some metal ions have emission-imprinting effects that cause strong quenching of fluorescence. Inherently quenched metal ions, such as Hg^2+^, Cu^2+^, Co^2+^, Ni^2+^, and Fe^3+^, can interfere with the fluorescence signals of sensors [34]. Therefore, LODs have been calculated as relatively high in copper detection studies based on fluorescence quenching using both porous silicon nanoparticles [35] and graphene oxide materials [36]. A fluorescence-quenching sensor system using an ion-imprinted polymer modified with carbon quantum dots was also similar. However, while the combined use of carbon quantum dots strengthened by ion imprinting was expected to have a synergistic effect, a higher LOD value (2.86 µM) was calculated compared to our ion-imprinted nanoparticles [37]. Lou et al. prepared a QCM sensor functionalized by SBA-15-SH mesoporous structures for the capture of Hg^2+^ ions from water. The selectivity of Hg^2+^ ions were measured in the presence of Cu^2+^ ions. However, a smaller coordination ability of copper ions than Hg^2+^ ions showed that the mesoporous silica material functionalized with single-bond -SH groups was not selective towards copper ions [38]. Finally, in this study a selective adsorption medium was prepared by means of Cu(II)-imprinted polymeric nanoparticles, and trace amounts of copper ions (42 nM) in aqueous media were detected with a highly sensitive QCM nanosensor system.

## 4. Conclusions

In this study, molecularly designed QCM nanosensors were applied for the real-time detection of Cu(II) ions in aqueous media. The interactions between the MAH monomers and Cu(II) ions were examined both computationally and experimentally. The structures of Cu(II)-MAH and Cu(II)-(MAH)_2_ were computed within four-fold and six-fold coordinations. Cu(II) showed distorted square planar geometry, with four-fold coordination in both structures. Due to the binding energies, Cu(II)-(MAH)_2_ was found to be more stable than the Cu(II)-(MAH) complex, which was in good agreement with the experimental results. Cu(II)-MIP nanoparticles were synthesized using these stable complexes according to the principle of the molecular-imprinting method. Cu(II)-MIP1 and MIP2 showed the change values of ΔF and Δm for Cu(II) solutions applied at different concentrations of nanoparticles. By binding the mass to the quartz surface, a decrease in crystalline oscillation frequency change, as well as a marked increase in mass awareness, were observed. With equilibrium isotherm data, the Cu(II)-MIP2 QCM nanosensor chip surface showed that the nanoparticle Cu(II)-binding sites were both heterogeneously and homogeneously distributed. The Cu(II)-MIP1 and MIP2 QCM nanosensors operated with high affinity and 99% accuracy in the linear concentration range of 0.15–1.57 µM. The LOD values were determined as 42 and 40.7 nM for the Cu(II)-MIP1 and Cu(II)-MIP2 QCM nanosensors, respectively. According to the selectivity results obtained from the Δm values, Cu(II)-MIP nanoparticles showed high selectivity for Cu(II) ions in the presence of competitive aqueous samples (Co (II), Ni (II), Fe (II), and Zn (II)) by preserving their ionic radius and shape memory. As a result of five repeated cycles, the Cu(II)-MIP-based QCM nanosensor could be used with high efficiency.

## Data Availability

Not applicable.

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
