# Peer review of "Molecularly Designed Ion-Imprinted Nanoparticles for Real-Time Sensing of Cu(II) Ions Using Quartz Crystal Microbalance"

_biomimetics, 2022, doi:10.3390/biomimetics7040191_

Round 1

Reviewer 1 Report

The manuscript predented a Cu(II) QCM sensor by ion imprinted polymer nanoparticles. The sensing  material design is of novelty and scentific value.I recommend its publication after a minor revision.

1. In the full text, superscript and subscript, capital and amall letter(Qcm) need to be checked carefully. The English writing should be polished, specially in stiff vocabulary.

2..Relative to metal ion detection by QCM, a suitable reference is suggested to cite or review, The real-time detection of trace-level Hg2+ in water by QCM loaded with thiol-functionalized SBA-15, Lou Huihui Zhang yuan Xiang Qun Xu Jiaqiang Li Hui Xu Pengcheng Li Xinxin, Sens Actuators B, 2012, 166, 246-252

3. The error bar is suggested to be added.

Author Response

We thank the reviewer for positive feedback and suggestions. We've highlighted the changes we've made to the text in red and added the details attachment file for each question.

Reviewer 2 Report

This manuscript presents the fabrication of QCM sensor for the detection of Cu(II) ions as wall as DFT study. The manuscript is well written. The results and discussion are sound. Therefore, it can be accepted for publication after minor revision as suggested below;

-       The references in this manuscript are out of date. Authors should review the current relative works in 2022.

-       The real photograph of the finish QCM sensor should be included in the manuscript.

-       In Figure 5, please improve the text labels. They are quite small.

Author Response

We would like to thank the reviewer for positive feedback and the detailed suggestions. Each question's changes have been marked in red in the text and are also shown at attachment. 

Reviewer 3 Report

please see attached

Author Response

Thank to the reviewer for suggestions and positive feedback. The whole text has been checked grammatically and changes are marked in red. The specific changes and adding suggested by the reviewer are also done. It can be find at attachment. 
